# Anti-Obesity Effect of *Auricularia delicate* Involves Intestinal-Microbiota-Mediated Oxidative Stress Regulation in High-Fat-Diet-Fed Mice

**DOI:** 10.3390/nu15040872

**Published:** 2023-02-08

**Authors:** Lanzhou Li, Siyu Zhai, Ruochen Wang, Fange Kong, Anhui Yang, Chunyue Wang, Han Yu, Yu Li, Di Wang

**Affiliations:** 1Engineering Research Center of Edible and Medicinal Fungi, Ministry of Education, Jilin Agricultural University, Changchun 130118, China; 2School of Life Sciences, Jilin University, Changchun 130012, China; 3Joint International Research Laboratory of Modern Agricultural Technology, Ministry of Education, Jilin Agricultural University, Changchun 130118, China; 4College of Agriculture, Jilin Agricultural University, Changchun 130118, China

**Keywords:** obesity, *Auricularia delicate*, intestinal microbiota, oxidative stress, Nrf2

## Abstract

*Auricularia delicate* (ADe), an edible fungus belonging to the family Auriculariaceae and order Auriculariales, possesses antimicrobial, hepatoprotective, and antioxidant effects. In this study, after systematic analysis of its composition, ADe was administered to high-fat-diet (HFD)-fed mice to investigate its anti-obesity effect. ADe significantly controlled body weight; alleviated hepatic steatosis and adipocyte hypertrophy; reduced aspartate aminotransferase, total cholesterol, insulin, and resistin; and increased adiponectin levels in HFD-fed mice serum. Based on intestinal microbiota and lipidomics analysis, ADe treatment regulated the composition and abundance of 49 intestinal microorganisms and influenced the abundance of 8 lipid species compared with HFD-fed mice. Based on a correlation analysis of the intestinal microbiota and lipids, *Coprococcus* showed significant negative associations with ceramide (d18:0 20:0+O), phosphatidylserine (39:4), sphingomyelin (d38:4), and zymosterol (20:2). Moreover, ADe treatment decreased the levels of ROS and MDA and increased the levels of Nrf2, HO-1, and three antioxidant enzymes in HFD-fed mice livers. Collectively, the anti-obesity effect of ADe involves the regulation of oxidative stress and is mediated by the intestinal microbiota. Hence, this study provides a reference for the application of ADe as a candidate food for obesity.

## 1. Introduction

Obesity results from an increased proportion of energy intake and consumption [1]. In China, the number of people who are overweight and obese has increased rapidly. According to the 2015–2019 national prevalence estimates, approximately 34.3% and 16.4% of adults were overweight and obese, respectively [2]. Obesity-related metabolic disorders, such as type 2 diabetes and nonalcoholic fatty liver disease, have become serious global health problems, affecting over 10% of the world’s population and causing almost 2.8 million deaths each year [3,4].

Bariatric surgery is an effective treatment for severe obesity [5], which may cause side effects such as tachycardia, peritonitis, malnutrition, and superior mesenteric vein thrombosis, accompanied by a long recovery period [6]. Several drugs (such as orlistat, liraglutide, and statins) have been approved for obesity or hyperlipidemia therapy. Unfortunately, orlistat, a pancreatic lipase inhibitor, can cause flatulence, cardiac disorders, and psychiatric disorders [7,8]. Liraglutide, a glucagon-like peptide-1 receptor agonist, can increase the risk of hypoglycemia, liver disease, acute pancreatitis, gallbladder disease, and biliary disease [9]. The use of statins (such as pravastatin and simvastatin [Sim]) may be accompanied by myalgia and rhabdomyolysis [10].

Diverse changes occur in the intestinal microbiota composition of patients with obesity and metabolic disorders. Sim treatment regulated the abundance of Firmicutes, Actinobacteria, and Bacteroidetes in phylum level [11]. The abundance of of bacteria of the genus *Coprococcus* in the feces of people with obesity has been shown to be significantly lower than that in lean people [12], and an increase in *Coprococcus* genus abundance helped alleviate insulin resistance and inhibit the development of obesity [13,14]. *Coprococcus* genus is the main producer of short-chain fatty acids (SCFAs), whose abundance could affect the level of visceral fat [15]. SCFAs serve as modulators of redox signaling [16], and can slow gastric emptying and promote triglyceride hydrolysis and fatty acid β-oxidation in the mitochondria by promoting hepatic autophagy [17].

Oxidative stress promotes the pathophysiology of obesity by changing the expression of mitochondrial activity regulators and the levels of inflammatory mediators associated with adipocytes and promoting adipogenesis and the differentiation of mature adipocytes [18]. Consequently, natural antioxidants have the potential to treat obesity [18]. The protein Nrf2, which is important for regulating the oxidative stress response, has become a potential new target against insulin resistance and subsequent obesity [19,20]. Oltipraz, a synthetic dithiolethione and an effective Nrf2 activator, can prevent increased weight and fat accumulation caused by a high-fat diet (HFD) in mice [21].

Recently, some natural products, especially mushrooms, which can antagonize oxidative stress and regulate blood lipid metabolism, have also become therapeutic candidates for obesity [18]. *Lentinula edodes*, a traditional edible mushroom, regulated lipid metabolism and inhibited lipid peroxidation in HFD-fed rats [22]. *Grifola frondosa*, an edible and medicinal mushroom, showed an anti-obesity effect in HFD-fed mice by regulating lipid metabolism via the ceramide pathway [23]. *Auricularia delicata* (ADe) (also known as Deer tripe mushroom in China and Uchina in India) belongs to the Auriculariaceae family and Auriculariales order. It grows mainly along regions near the equator [24]. As a potential nutritious food source with pharmaceutical functions, ADe has been used in some regions of Africa and Eastern Asia, particularly in China and India [25], to treat various gastrointestinal and liver diseases [26]. Recent research has found that ADe extracts showed antimicrobial, hepatoprotective, and antioxidant effects [25]. Polysaccharides from ADe effectively scavenged -OH and O_2_^-^ in vitro, indicating its antioxidant properties [27]. However, few studies have examined the regulatory effects of ADe on obesity and elucidated its underlying mechanisms [25].

In this study, a diet-induced obesity (DIO) mouse model was used to investigate the lipid metabolism regulatory effect of ADe on obesity. Based on intestinal microbiota and lipidome analysis, ADe showed anti-obesity effects via intestinal-microbiota-mediated oxidative stress regulation. These results are helpful for the development of ADe as a functional food for obesity.

## 2. Materials and Methods

### 2.1. ADe Component Detection

ADe fruiting bodies that were planted and collected at the Engineering Research Center of the Chinese Ministry of Education for Edible and Medicinal Fungi, Jilin Agricultural University, were identified by Prof. Yu Li from Jilin Agriculture University. The fruiting bodies were ground using a blender (to a size less than 74 μm) and stored dry until use.

#### 2.1.1. Nutritional Composition Analysis

The nutritional composition, including the levels of total sugar, reducing sugar, crude protein, total ash, crude fat, crude fiber, total flavonoids, total triterpenoids, mannitol, total phenol, total sterol, and vitamins in ADe was systematically measured as described in our previous study [28]. Total saponins and total alkaloids were assessed via UV spectrophotometry [29,30].

#### 2.1.2. Mineral Analysis

ADe was mixed with nitric acid (2:5, *w*:*v*) and digested at 100, 140, 160, and 180 °C for 3 min, respectively, and 190 °C for 15 min. The calcium (Ca), magnesium (Mg), potassium (K), and sodium (Na) content was analyzed using inductively coupled plasma optical emission spectrometry (Optima 8000; PerkinElmer, Waltham, MA, USA). The levels of arsenic (As), cadmium (Cd), copper (Cu), chromium (Cr), iron (Fe), lead (Pb), mercury (Hg), manganese (Mn), selenium (Se), and zinc (Zn) were analyzed using inductively coupled plasma mass spectrometry (iCAPQ; Thermo Fisher Scientific, Waltham, MA, USA) [23].

### 2.2. Animal Experiments and Agent Administration Protocol

The animal experimental protocol was approved by the Ethical Committee of Animal Research of Jilin University (SY201909007, approval date 10 September 2019). Thirty-six healthy male C57BL/6 mice (5–6 weeks old) (SCXK (Su) 2018-0008) were obtained from GemPharmatech Co., Ltd. (Jiangsu, China). The mice were housed in a controlled environment at a temperature of 23 ± 1 °C with 50 ± 10% humidity and a 12-h/12 h light-dark cycle with free access to water and food. Twelve mice were fed a normal chow diet (NCD, #D12450B (# represents the Article Number); 10% kcal fat, 20% kcal protein, and 70% kcal carbohydrate) (NCD: corn, soybean meal, wheat bran, wheat flour, soybean oil, fish meal, calcium hydrogen phosphate, stone powder, salt, choline chloride, lysine, multiple vitamins, and mineral elements) (Xiao Shu You Tai Biotechnology Co., Ltd., Beijing, China), and twenty-four mice were fed a HFD (#D12492; 60% kcal fat, 20% kcal protein, and 20% kcal carbohydrate) (HFD: add lard, egg yolk powder, and cholesterol to NCD) to establish the diet-induced obesity (DIO) model for 18 weeks. Simvastatin (Sim) was selected as a positive drug to study the effect of ADe on blood lipids [31]. ADe fruiting body was dried and ground, and then mixed with normal saline to 80 or 160 mg/mL, and Sim mixed with normal saline to 0.6 mg/mL. Normal saline was used as vehicle for controls. From the ninth week, the HFD-fed mice were randomly divided into four groups (n = 6 per group), including vehicle-treated HFD-fed mice (orally administered 5 mL/kg normal saline once a day), Sim-treated HFD-fed mice (orally administered 3 mg/kg Sim once a day), and 400 or 800 mg/kg ADe-treated HFD-fed mice (orally administered 400 or 800 mg/kg ADe once a day). The NCD-fed mice were also randomly divided into two groups (n = 6 per group), including vehicle-treated NCD-fed mice (orally administered 5 mL/kg normal saline once a day) and ADe-treated NCD-fed mice (orally administered 400 mg/kg ADe once a day). After the last treatment, all mice were fasted for 12 h, and the blood glucose levels were measured using a fast blood glucose meter [32]. After sampling blood from the caudal veins, the mice were euthanized via CO_2_ inhalation. The cecum contents were collected under aseptic conditions and stored at −80 °C for intestinal microbiota analysis. The epididymal white adipose tissue (eWAT), perirenal white adipose tissue (pWAT), inguinal white adipose tissue (iWAT), liver, spleen, kidney, heart, and pancreas were collected and weighed. The tissue coefficients were calculated as follows:(1)$$\mathrm{Tissue}\;\mathrm{coefficients}\;\left(\%\right)=\mathrm{tissue}\;\mathrm{weight}\;(g)/\mathrm{body}\;\mathrm{weight}\;(g)\;\times \;100$$

### 2.3. Biochemical Analysis

Liver tissues were homogenized with physiological saline, centrifuged twice at 3500 r/min for 10 min each time, and the protein concentrations were determined using a bicinchoninic acid protein assay kit (#23227; Thermo Fisher Scientific). The levels of high-density lipoprotein cholesterol (HDL-C, #KT2827-A), total cholesterol (#KT30043-A), low-density lipoprotein cholesterol (LDL-C, #KT2797-A), aspartate aminotransferase (AST, #KT2856-A), glycosylated hemoglobin A1c (GHbA1c, #KT2511-A), insulin (#KT2579-A), adiponectin (#KT2547-A), resistin (#KT2552-A), ROS (#KT9261-A), malondialdehyde (MDA, #KT9264-A), glutathione peroxidase (GSH-Px, #KT2755-A), and catalase (CAT, #KT2833-A) were determined using enzyme-linked immunosorbent assay kits (Jiangsu Kete Biotechnology Co., Ltd., Jiangsu, China) according to the manufacturer’s instruction manuals.

### 2.4. Histopathological Analysis and Immunohistochemical Examination

The iWAT, eWAT, pWAT, liver, and pancreas were fixed in 10% formalin buffer for 48 h, embedded in paraffin, and sliced into 5 μm thick sections. For Oil Red O staining, the sections of liver were stained with 3.7 mM Oil Red O dissolved and then counterstained with hematoxylin as in a previous study [33]. For hematoxylin and eosin (H&E) staining, the sections were stained with hematoxylin for 5 min and eosin for 3 min at 25 °C as in our previous study [23]. For immunohistochemistry staining, the sections of pancreas were blocked with 5% bovine serum albumin (Gen-view Scientific, Galveston, TX, USA), then incubated with anti-insulin antibody (#ab181547, 1:2000 dilution) (Abcam, Cambridge, MA, USA) at 4 °C overnight. The sections were then incubated with horseradish peroxidase (HRP)-labeled goat anti-rabbit antibody (#E-AB-1003) (Elabscience, Wuhan, China) at 4 °C for 2 h. Color development was performed using a DAB Kit (#34065, Thermo Fisher, Carlsbad, CA, USA), and hematoxylin was used for counter staining.

The stained slides were observed under a light microscope (Olympus, Tokyo, Japan) and photographed.

### 2.5. Intestinal Microbiota Analysis

Cecal contents were randomly collected from vehicle-treated NCD-fed mice (n = 3), vehicle-treated HFD-fed mice (n = 3), and 400 mg/kg ADe-treated HFD-fed mice (n = 4) for routine microbiota total DNA extraction and stored at −80 °C. Microbial DNA was extracted, the bacterial 16S rRNA gene V3–V4 region was amplified via PCR, and paired-end 2 × 250 bp sequencing was performed using an Illumina MiSeq platform with MiSeq Reagent Kit v3 at Shanghai Personal Biotechnology Co., Ltd. (Shanghai, China) as in our previous study [23]. The bacteria sequences were uploaded to the NCBI Sequence Read Archive under the accession number PRJNA842506 (https://www.ncbi.nlm.nih.gov/sra/PRJNA842506/, accessed on 26 May 2022.). The results were analyzed as previously described [34].

### 2.6. Lipidome Analysis

Serum samples from 400 mg/kg ADe-treated HFD-fed mice, vehicle-treated HFD-fed mice, and vehicle-treated NCD-fed mice (n = 3) were randomly collected for lipidome analysis. The samples were analyzed via LC-MS/ESI-MSn using a Thermo Ultimate 3000 system equipped with an ACQUITY UPLC BEH C18 column and a Thermo Q Exactive Focus mass spectrometer (Thermo Fisher Scientific), as described in our previous study [23]. Significant statistical differences in the metabolites (*p* ≤ 0.05) and variable importance in projection values ≥1.0 were regarded as the standard values for differential lipids to filter out biomarkers.

### 2.7. Western Blotting

As in our previous study, the liver samples were lysed using a radioimmunoprecipitation assay lysis buffer. After protein content analysis, 40 µg isolated proteins were separated using 10–12% sodium dodecyl sulfate-polyacrylamide gel electrophoresis and transferred onto 0.45-μm polyvinylidene difluoride membranes (Merck, Darmstadt, Germany). The membranes were then blocked with 5% bovine serum albumin at 4 °C for 4 h and then incubated with primary antibodies, including CAT (#ab209211, 1:2000 dilution), Nrf2 (#ab92946, 1:1000 dilution), glyceraldehyde-3-phosphate dehydrogenase (GAPDH) (#ab181602, 1:1000 dilution) (Abcam, Cambridge, MA, USA), heme oxygenase 1 (HO-1, #A19062, 1:1000 dilution), and superoxide dismutase 1 (SOD-1, #A12537, 1:1000 dilution) (Abclonal, Wuhan, China) overnight at 4 °C. The horseradish peroxidase (HRP)-conjugated secondary antibodies (#E-AB-1001 and #E-AB-1003) (Elabscience, Wuhan, China) were used to incubate with the membranes at 4 °C for 4 h. Protein bands were developed using an enhanced chemiluminescence detection kit (Merck, Darmstadt, Germany) and visualized using a Tanon 5200 gel imaging system (Tanon Science & Technology Co., Ltd., Shanghai, China). The pixel density was measured using ImageJ software (National Institutes of Health, Bethesda, MD, USA) [23].

### 2.8. Statistical Analysis

All values are expressed as the mean ± S.E.M. One-way analysis of variance (ANOVA) was performed to detect statistical significance using BONC DSS Statistics 25 software (Business-intelligence of Oriental Nations Co., Ltd., Beijing, China). *p* values < 0.05 were considered statistically significant.

## 3. Results

### 3.1. Composition of ADe

The main components of ADe were 54.6% total sugar, 12.4% crude fiber, 8.18% crude protein, 3.8% total ash, 2.6% crude fat, 2.58% reducing sugar, 0.13% total phenol, 0.1% total saponin, 0.385% total sterol, 81.1 × 10^−3^% vitamin C, 36.1 × 10^−5^% vitamin E, 19.3 × 10^−5^% vitamin B3, 5.6 × 10^−4^% vitamin D2, 55.5 × 10^−5^% vitamin D3, and 84.7 × 10^−5^% pyridoxal (Table 1). Total flavonoids and vitamins A, B1, B2, pyridoxine, and pyridoxamine were not detected (Table 1). Hg and Cd were not detected among the 14 minerals analyzed (Table 1).

### 3.2. The Anti-Obesity Effect of ADe

Consistent with previous studies, obvious obesity-related lesions were observed in the HDF-fed mice [23]. Similar to Sim treatment, from the fourth week, ADe treatment significantly reduced the body weight of HFD-fed mice (*p* < 0.05) (Appendix A) and effectively inhibited the decrease in spleen, kidney, and heart indices (*p* < 0.05) (Appendix A). ADe treatment induced over 10.27% (*p* < 0.05), over 18.92% (*p* < 0.05) and over 26.24% (*p* < 0.01) reductions in the indices of eWAT, pWAT, and iWAT, respectively, in HDF-fed mice (Figure 1A). In NCD-fed mice, ADe reduced the body weight (*p* < 0.05) (Appendix A), suppressed the indices of eWAT and pWAT (Figure 1A), and increased the index of the kidney (*p* < 0.01) (Appendix A). A remarkable increase in food and water intake was noted (*p* < 0.01) in vehicle-treated HFD-fed mice, which was decreased by ADe and Sim treatment (*p* < 0.01) (Figure 1B). ADe alone did not change the food intakes of NCD-fed mice (*p* > 0.05) (Figure 1B), suggesting no relationship between the anti-obesity and diet inhibition effects of ADe. The adipocytes in the eWAT, pWAT, and iWAT of HFD-fed mice were larger than those in NCD-fed mice, which were reversed by ADe treatment (Figure 1C).

### 3.3. ADe Treatment Regulated Blood Lipids and Alleviated Hepatic Steatosis

The liver is a hub for fatty acid synthesis and lipid circulation [35]. According to Oil Red O and H&E staining results, ADe treatment strongly promoted the maintenance of liver cell structure and reduced fat vesicles and lipids in HFD-fed mice liver (Figure 2A,B). Compared with HFD-fed mice, ADe reduced the liver indices by over 45.48% (*p* < 0.001) (Figure 2C), serum AST levels by over 21.62% (*p* < 0.05) (Figure 2D), serum total cholesterol by 25.65% (at a 400 mg/kg dose) (*p* < 0.001) (Figure 2E), serum LDL-C by over 14.00% (*p* < 0.05) (Figure 2F), and increased serum HDL-C by over 24.96% (*p* < 0.01) (Figure 2G). Sim treatment significantly decreased the liver index, serum AST, serum LDL-C levels, serum total cholesterol, and increased serum HDL-C levels in HFD-fed mice (*p* < 0.01) (Figure 2C–G).

### 3.4. ADe Protected Pancreatic Function

Insulin resistance can lead to hyperglycemia, promote lipid synthesis, and induce fat accumulation in the liver [36]. ADe treatment ameliorated the vacuolation and degranulation of pancreatic cells as analyzed via H&E staining (Figure 3A). Moreover, immunohistochemistry staining showed that ADe suppressed the expression of insulin in the pancreas of HFD-fed mice (Figure 3B). Compared to HFD-fed mice, ADe treatment increased the pancreas index by over 24.28% (*p* < 0.01) (Figure 3C) and serum adiponectin by over 58.08% (*p* < 0.001) (Figure 3H), and decreased blood glucose by over 9.26% (*p* < 0.001) (Figure 3D), serum GHbA1c by over 17.20% (*p* < 0.01) (Figure 3E), serum insulin by 14.24% (at a 800 mg/kg dose) (*p* < 0.01) (Figure 3F), and serum resistin by over 24.96% (*p* < 0.01) (Figure 3G).

### 3.5. ADe Regulated the Intestinal Microbiota

According to the Venn results, the common number of amplicon sequence variants (ASVs) between the three groups was 173, showing a significant difference in microbiota composition (Figure 4A). The rank abundance curve showed that the evenness of the intestinal microbiota of HFD-fed mice was lower than that of NCD-fed mice, and ADe treatment improved the intestinal microbiota evenness in HFD-fed mice (Figure 4B). Two-dimensional unweighted UniFrac-based PCoA ordination showed that microbial community composition could be easily discriminated between the three groups (Figure 4C). Compared to vehicle-treated NCD-fed mice, observed species, Simpson, Shannon, Pielou’s e, and Faith’s PD were significantly reduced in HFD-fed mice (*p* < 0.05) (Figure 4D), which explained the reduced richness, evenness, and diversity of the intestinal microbiota in mice administered with an HFD. According to the heatmap created, ADe increased the abundance of 19 prevalent bacterial genera and reduced the abundance of 9 prevalent bacterial genera in HFD-fed mice (Figure 4E).

LEfSe analysis was used to analyze different biomarkers in the cecal content at all classification levels (Figure 4F). There were 49 significantly different nodes (including 19 bacterial genera) between the 3 groups (*p* < 0.05, LDA > 2) (Figure 4F). *Odoribacter*, *Clostridium* (Erysipelotrichaceae family), *Coprococcus*, *Butyricicoccus*, and *Akkermansia* genera demonstrated the highest abundance in NCD-fed mice; *Ruminococcus*, *Allobaculum*, and *Clostridium* (Lachnospiraceae and Clostridiaceae families) genera showed the highest abundance in HFD-fed mice; and *Bacteroides*, *Lactococcus*, *Dehalobacterium*, *Dorea*, *Anaerotruncus*, *Oscillospira*, *Sphingomonas*, *Burkholderia*, *Cupriavidus*, and *Ralstonia* genera were most abundant in ADe-treated HFD-fed mice (Figure 4F).

### 3.6. ADe Regulated the Metabolism of Lipids Related to the Nrf2/HO-1 Pathway

Serum lipidomics analysis revealed ten markedly different lipid species (Figure 5A). Among them, an increase in Cer (d18:0 20:0+O), CL (88:11), DG (18:1 18:1), PS (39:4), PS (18:0 22:6), SM (d38:4), and ZyE (20:2) and decreased content of PE (20:1e 18:2) were observed in HFD-fed mice compared with NCD-fed mice (Figure 5B,C); all of these were reversed upon ADe treatment in HFD-fed mice (Figure 5B,C). The correlation heatmap of these eight lipid species showed a positive association between Cer (d18:0 20:0+O) and SM (d38:4) levels, CL (88:11) and DG (18:1 18:1) levels, and PS (39:4) and ZyE (20:2) levels (Figure 5D).

The correlation heatmap of 8 differential lipids and 19 biomarker intestinal microorganisms showed that *Ruminococcus*, *Coprococcus*, and *Butyricicoccus* genera were significantly associated with the abundance of some lipids (Figure 6A). Among these, *Coprococcus* genus had the most significant negative association with Cer (d18:0 20:0+O) (*p* < 0.01), PS (39:4) (*p* < 0.01), SM (d38:4) (*p* < 0.05), and ZyE (20:2) (*p* < 0.01) (Figure 6A). Compared to vehicle-treated HFD-fed mice, ADe treatment reduced *Ruminococcus* genus and increased *Coprococcus* and *Butyricicoccus* genera in HFD-fed mice (Appendix A).

Furthermore, ADe treatment significantly reduced ROS levels by over 33.98% (*p* < 0.001) (Figure 6B) and MDA by over 31.09% (*p* < 0.01) (Figure 6C), and increased the levels of GSH-Px by over 53.60% (*p* < 0.05) (Figure 6D) and CAT by over 35.17% (*p* < 0.01) (Figure 6E) in HFD-fed mice liver. Western blotting showed that ADe significantly increased the levels of Nrf2 by over 76.18% (*p* < 0.001), HO-1 by over 32.21% (*p* < 0.001), SOD-1 by over 19.75% (*p* < 0.001), and CAT by over 69.70% (*p* < 0.001) in HFD-fed mice liver (Figure 6F).

## 4. Discussion

In the present study, the composition of ADe was systematically determined. Total sugar (54.6%) was the most abundant component in ADe, and a high abundance of crude fiber (12.4%) and vitamin C (811 mg/kg) was also observed. High dietary fiber intake can reduce the body weight of obesity patients, reducing their risk of having coronary heart disease, diabetes, and gastrointestinal diseases [37]. Vitamin C has a strong antioxidant effect which can effectively inhibit female abdominal obesity [38]. Our results showed that ADe treatment strongly reduced body weight and fat accumulation, protected the structure and function of the liver and pancreas, and partially regulated lipid levels via the regulation of the intestinal microbiota and inhibition of oxidative stress in HFD-fed mice.

ADe treatment significantly suppressed the AST and total cholesterol levels in HFD-fed mice. Both AST and total cholesterol can serve as evaluation indices in patients with obesity, reflecting the degree of hepatocyte damage and increase in adipocytes [39,40]. In patients who are overweight or obese, adipose tissue dysfunction combined with glycotoxicity and lipotoxicity can seriously affect the pancreas and liver, leading to pancreatic β-cell damage and hepatic insulin resistance [41,42]. During insulin resistance, the insensitivity of cells to insulin forces more insulin production, leading to hyperinsulinemia and type 2 diabetes [43]. ADe has been previously shown to suppress GHbA1c, insulin, and resistin levels, and enhance adiponectin levels. A high level of circulating resistin is the main cause of insulin resistance [44]. GHbA1c levels reflect the plasma glucose status 2–3 months before measurement and can be used for evaluating and screening for insulin resistance [45]. Adiponectin, secreted by the adipose tissue, can reduce serum triglyceride levels by enhancing the catabolism of triglyceride-rich lipoproteins [46]. Hence, the ADe anti-obesity effect is related to regulate insulin resistance, which is also affected by the intestinal microbiota [47].

ADe substantially changed the abundance and composition and improved the diversity and evenness of intestinal microbiota in HFD-fed mice, including *Bacteroides*, *Lactococcus*, *Butyricicoccus*, and *Coprococcus* genera. The use of probiotics and antibiotics helps reshape the metabolic characteristics of patients with obesity [48]. *Bacteroides* genus, significantly reduced in obese mice, can help reduce weight gain [49]. The increase of *Butyricoccus* genus, a main producer of butyrate [50], can help to improve metabolic risk via decreasing LDL-C and total cholesterol [51]. *Lactococcus* and *Coprococcus* genera are two main producers of SCFAs [52], both can regulate adipose tissue metabolism, alleviate insulin resistance [13,53], and help prevent oxidative stress via the activation of Nrf2 [16]. Therefore, changes in the abundance of intestinal microbiota could influence lipid composition. *Sargassum fusiforme* fucoidan increased the abundance of tauroursodeoxycholic acid related to *Clostridium* genus and inhibited FXR/SHP signaling to reduce Cer synthesis [54]. A high dose of xylan-oligosaccharides increased the levels of *Bifidobacterium*, *Lachnospiraceae_NK4A136_group*, and *Roseburia* genera and decreased the levels of Cer in HFD-fed mice [55]. *Lactobacillus plantarum* significantly reduced SM levels [56], and *Bifidobacterium longum* subsp. *longum* intervention reduced PS levels in HFD-fed mice [57]. Similarly, ADe can affect the lipid composition (levels of Cer, PS, SM, and ZyE) and is negatively correlated with the content of *Coprococcus* genus.

SM and Cer are related to metabolic syndrome and liver function in obesity patients [58]. SM increases oxidative stress by decreasing calcium ATPase activity in the plasma membrane [59,60]. Obesity-induced oxidative stress in adipose tissue can be controlled by ROS [61], which further influences the levels of lipid peroxidation products, such as MDA, by evaluating the degree of lipid oxidative damage [62]. Antioxidant enzymes, such as GSH-Px, CAT, and SOD, can effectively scavenge ROS and prevent oxidative stress [19,20]. Nrf2 has become a potential anti-obesity target due to its antioxidant and anti-inflammatory effects [63]. As previously reported, *Grifola frondosa* extract can reduce fat accumulation and inhibit oxidative damage, which is partially related to the Nrf2 signaling pathway [64]. ADe treatment effectively reduced the levels of pro-oxidative factors and increased the levels of anti-oxidative factors, which was proven to be related to the enhancement of Nrf2 signaling in HFD-fed mice liver. The present data confirm the crucial role of Nrf2-related oxidative stress during the ADe-mediated suppression of the development of obesity.

The present study has certain limitations. ADe contains various components; however, the main active component of ADe in obesity remains unclear. The sample size of this study is limited, and it needs to be further verified by a larger-sample-size study. Moreover, ADe regulated other intestinal microorganisms, and the role of these microorganisms in the anti-obesity effect requires further investigation.

## 5. Conclusions

ADe treatment can inhibit the development of obesity in HFD-fed mice, which is related to its regulation of the intestinal microbiota, and inhibition of oxidative stress via Nrf2 signaling. This study provides a reference for development of ADe as a candidate food for obesity.

## Figures and Tables

**Figure 1 nutrients-15-00872-f001:**
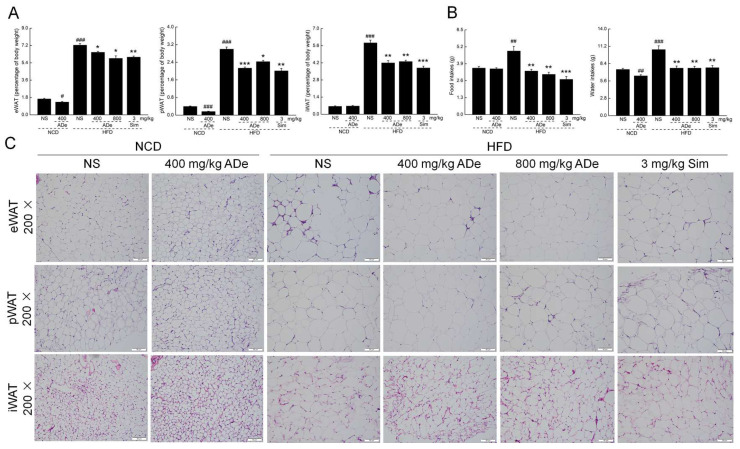
Effect of ADe on the WATs and food and water intake. (**A**) ADe suppressed the eWAT, pWAT, and iWAT indices in DIO mice (n = 6). (**B**) ADe decreased the food and water intakes in DIO mice (n = 6). (**C**) H&E staining of the eWAT, iWAT, and pWAT (200×; scale bar: 50 μm). ^#^
*p* < 0.05, ^##^
*p* < 0.01, and ^###^
*p* < 0.001 versus NCD-fed mice; * *p* < 0.05, ** *p* < 0.01, and *** *p* < 0.001 versus HFD-fed mice.

**Figure 2 nutrients-15-00872-f002:**
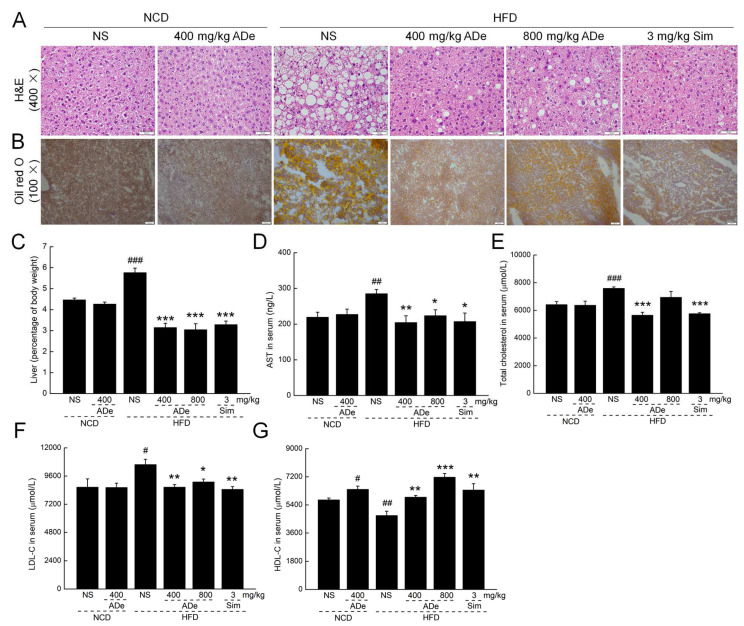
Effect of ADe treatment on the liver function. Histopathological analysis of the mouse livers after (**A**) H&E staining (200×; scale bar: 50 μm) and (**B**) Oil Red O staining (100×; scale bar: 100 μm). ADe decreased (**C**) the liver index, the serum levels of (**D**) AST, (**E**) total cholesterol, and (**F**) LDL-C, and increased the serum levels of (**G**) HDL-C (n = 6). ^#^
*p* < 0.05, ^##^
*p* < 0.01, and ^###^
*p* < 0.001 versus NCD-fed mice; * *p* < 0.05, ** *p* < 0.01, and *** *p* < 0.001 versus HFD-fed mice.

**Figure 3 nutrients-15-00872-f003:**
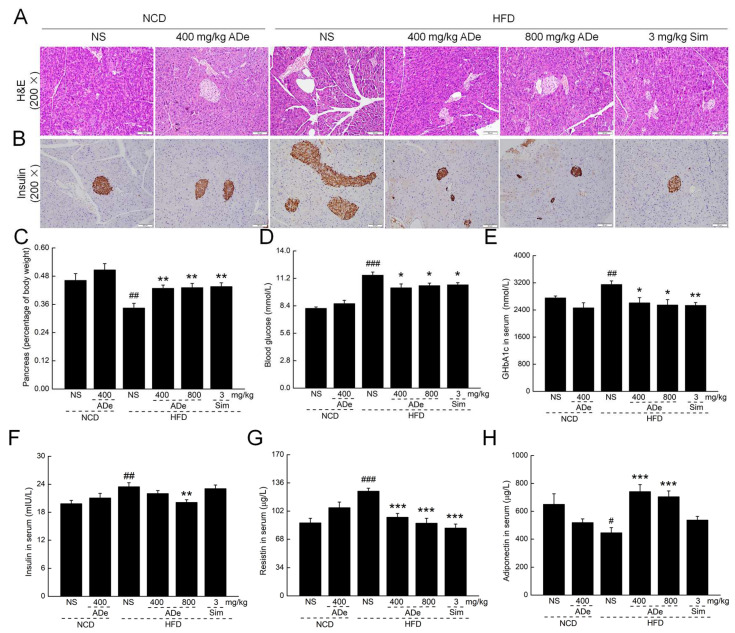
Effect of ADe treatment on pancreatic function. (**A**) Histopathological analysis of the pancreas using H&E staining (200×; scale bar: 50 μm). (**B**) ADe reduced pancreatic insulin levels (200×; scale bar: 50 μm) as analyzed via immunohistochemistry. ADe treatment increased (**C**) the pancreas index, decreased (**D**) the blood glucose, (**E**) GHbA1c, (**F**) insulin, and (**G**) resistin levels, and increased serum (**H**) adiponectin levels (n = 6). ^#^
*p* < 0.05, ^##^
*p* < 0.01, and ^###^
*p* < 0.001 versus NCD-fed mice; * *p* < 0.05, ** *p* < 0.01, and *** *p* < 0.001 versus HFD-fed mice.

**Figure 4 nutrients-15-00872-f004:**
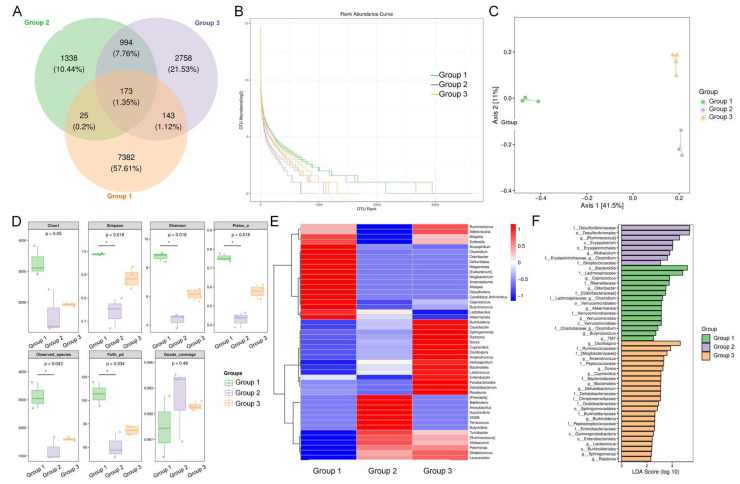
ADe treatment regulated the intestinal microbiota. (**A**) Venn diagram. (**B**) Rank abundance curve. (**C**) Two-dimensional unweighted UniFrac-based PCoA ordination of the mice intestinal microbiota. A closer projection distance between two points on the coordinate axis indicates a higher similarity in the community composition of the two samples in the corresponding dimensions. (**D**) Alpha diversity analysis of the experimental groups. Data are shown as the mean ± S.E.M. and statistical significance was determined via the Kruskal–Wallis test. * *p* < 0.05 versus NCD-fed mice. (**E**) Heatmap of the top 50 bacterial genera using the unweighted UniFrac distance from the cecal content samples. (**F**) Histogram of LDA effect values of marker species based on LEfSe analysis (LDA > 2 and *p* < 0.05; determined via the Kruskal–Wallis test) of the intestinal microbiota. Group 1: NCD-fed mice, Group 2: HFD-fed mice, Group 3: ADe-treated HFD-fed mice.

**Figure 5 nutrients-15-00872-f005:**
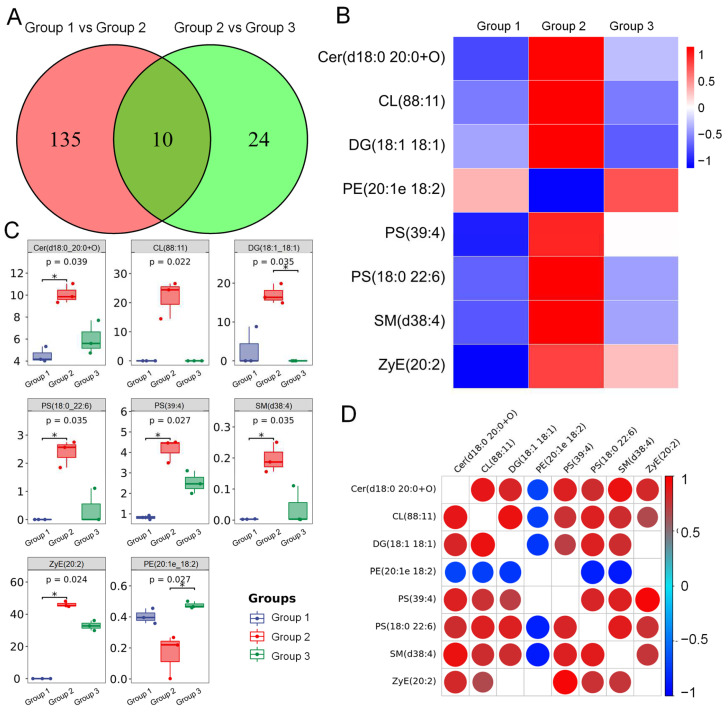
ADe treatment regulated the lipid metabolites in HFD-fed mice. (**A**) Venn diagram of significantly altered metabolites. (**B**) Heatmap and (**C**) box-plot of eight significantly altered metabolites in ADe-treated HFD-fed mice (n = 3). * *p* < 0.05 versus HFD-fed mice. (**D**) Correlation heatmap of eight significantly altered metabolites based on the Pearson correlation. Red means positive correlation and blue means negative correlation. The larger the circle, the stronger the significance. Cer, Ceramide; CL, Cardiolipin; DG, Diacylglycerol; PE, Phosphatidylethanola; PS, Phosphatidylserine; SM, Sphingomyelin; ZyE, Zymosterol. Group 1: NCD-fed mice, Group 2: HFD-fed mice, Group 3: ADe-treated HFD-fed mice.

**Figure 6 nutrients-15-00872-f006:**
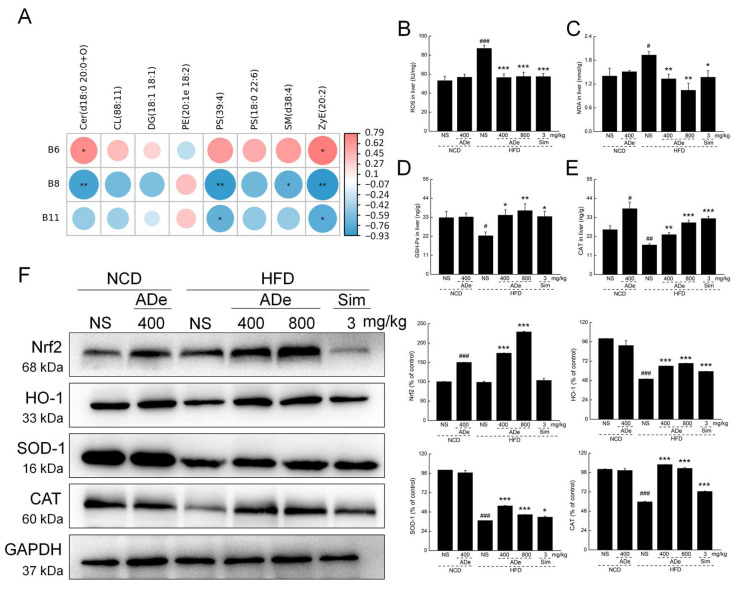
ADe treatment regulated oxidative stress levels. (**A**) Correlation heatmap of 8 significantly altered lipids and 19 biomarker intestinal microorganisms at the genus level based on the Pearson correlation coefficient with a *p*-value < 0.05 and R value > 0.75. B6: *Ruminococcus*, B8: *Coprococcus*, B11: *Butyricicoccus*. ADe decreased (**B**) ROS and (**C**) MDA levels and increased (**D**) GSH-Px and (**E**) CAT levels in HFD-fed mice liver. (**F**) The expression of Nrf2, HO-1, CAT, and SOD-1 in the liver were detected via Western blotting. The quantitative data of the protein expression levels were normalized to the expression of GAPDH and shown as a percentage of the corresponding relative intensity in the vehicle-treated NCD-fed mice. ^#^
*p* < 0.05, ^##^
*p* < 0.01, and ^###^
*p* < 0.001 versus NCD-fed mice; * *p* < 0.05, ** *p* < 0.01, and *** *p* < 0.001 versus vehicle-treated HFD-fed mice.

**Table 1 nutrients-15-00872-t001:** The main components of ADe.

	Name	Contents (%)	Name	Contents (%)
General nutritional compositions	Total sugar	54.6	Total triterpenoids (×10^−1^)	0.2
	Reducing sugar (×10^−1^)	25.8	Mannitol (×10^−2^)	0.6
	Crude protein (×10^−1^)	81.8	Total phenol (×10^−1^)	1.3
	Total ash	3.8	Total saponins	0.1
	Crude fat	2.6	Total alkaloids (×10^−2^)	1.2
	Crude fiber	12.4	Total sterol (×10^−2^)	38.5
	Total flavonoids	UD^*a*^		
Minerals	Calcium (Ca) (×10^−2^)	22.8	Magnesium (Mg) (×10^−2^)	16.8
	Iron (Fe) (×10^−2^)	1.6	Potassium (K) (×10^−1^)	11.5
	Zinc (Zn) (×10^−5^)	93.6	Sodium (Na) (×10^−6^)	6.8
	Selenium (Se) (×10^−6^)	10.3	Manganese (Mn) (×10^−5^)	57.5
	Lead (Pb) (×10^−6^)	56.5	Cadmium (Cd)	UD^*b*^
	Arsenic (As) (×10^−4^)	10.6	Copper (Cu) (×10^−5^)	29.8
	Mercury (Hg)	UD^*b*^	Chromium (Cr) (×10^−4^)	23.9
Vitamins	Vitamin A	UD^*c*^	Pyridoxine	UD^*f*^
	Vitamin E (×10^−5^)	36.1	Pyridoxal (×10^−5^)	84.7
	Vitamin C (×10^−3^)	81.1	Pyridoxamine	UD^*f*^
	Vitamin B1	UD^*d*^	Vitamin D2 (×10^−4^)	5.6
	Vitamin B2	UD^*e*^	Vitamin D3 (×10^−5^)	55.5
	Vitamin B3 (×10^−4^)	19.3		

UD: undetected. UD *^a^*: the detection limit was 500 mg/kg. UD *^b^*: the detection limit was 0.05 mg/kg. UD *^c^*: the detection limit was 3 mg/kg. UD *^d^*: the detection limit was 0.32 mg/kg. UD *^e^*: the detection limit was 0.13 mg/kg. UD *^f^*: the detection limit was 0.50 mg/kg.

## Data Availability

The bacteria sequences were uploaded to the NCBI Sequence Read Archive under the accession number PRJNA842506 (https://www.ncbi.nlm.nih.gov/sra/PRJNA842506/, accessed on 26 May 2022.).

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
