# Peer review of "Anti-Obesity Effect of Auricularia delicate Involves Intestinal-Microbiota-Mediated Oxidative Stress Regulation in High-Fat-Diet-Fed Mice"

_nutrients, 2023, doi:10.3390/nu15040872_

Round 1

Reviewer 1 Report

The manuscript entitled: “Anti-obesity effect of Auricularia delicate involve intestinal microbiota-mediated oxidative stress regulation in high-fat diet-fed mice” is an interesting work that can contribute to the scientific knowledge of this edible fungus health benefits. The authors presented data to support that Ade controlled body weight; alleviated hepatic steatosis and adipocyte hypertrophy; reduced aspartate aminotransferase, total cholesterol, insulin, and resistin, and increased adiponectin levels in HFD-fed mice serum. In addition, lipidomics and microbiota analyses showed that ADe treatment regulated the composition and abundance of 49 intestinal microorganisms and influenced the abundance of eight lipid species compared with HFD-fed mice. Furthermore, a correlation analysis of the intestinal microbiota and lipids, showed a negative significant correlation between Coprococcus with ceramide (d18:0 20:0+O), phosphatidylserine (39:4), sphingomyelin (d38:4), and zymosterol (20:2). Ade also decreased the levels of ROS and MDA and increased the levels of Nrf2, HO-1, and three antioxidant enzymes in HFD-fed mice liver. However, major review is needed.

Methods

1.     ADe-treated HFD-fed mice (were orally administered 400 or 800 mg/kg ADe once a day). There is not information about the preparation of Ade to feed the mice and the form of delivery. Authors should provide a complete description about ADe preparation or processing before incorporation into diet, or any other channel used to feed animals. What vehicle was used for controls?.

2.     Supplementary Figure S1, does not clearly describe the meaning of Group1, Group 2 and Group3 as has been done in Figure 4.

3.     Data from liver biomarkers shows that 800 mg DAe supplementation was less effective that 400 in the HFD mice. What is the explanation for this?

Results

4.     Serum lipidomics analysis showed significant differences in lipid species among the three experimental groups that were analyzed. Figure 5A is a Venn diagram showing only 2 circles instead of three to clearly illustrate the number of lipids that were common to each pair of groups and/or to the three groups as well as to only one group.  Moreover, full name of biomarkers shown in box plots should be added to figure caption. Similarly, for the correlation heatmap (Fig. 5D), an explanation of color interpretation is needed. I see that ZyE(20:2) is highly correlated with itself (***) in red color (red usually is negative correlation wile blue is positive), which wouldn’t make sense. Spearman correlation is recommended with ellipses indicating high or lower correlation according to shape.

Discussion

5.     Authors mention ADe contains various components (as presented in Table 1); however, the main active component of ADe in obesity remains unclear. It all depends on what was fed to animals, if an ADe extract was used, authors could have analyzed the extract to make an statement regarding the bioactive components modulating levels of biomarkers for obesity and microbiota analyzed in this study.

Reviewer 2 Report

More information should be added about the study design and methodology used. The conclusion should answer the objective (to investigate ADe anti-obesity effect). Since this study has been carried out in animals, any extrapolation to humans should be avoided.

Introduction

-       lines 49-50: Firmicutes, Actinobacteria and Bacteroidetes should be written in italics. Check the bacterial nomenclature throughout the manuscript.

-       Authors should indicate whether they are writing about genus, species, or strains. For example, line 50, instead of “The abundance of Coprococcus” it should be written “The abundance of bacteria of the genus Coprococcus”. 

-       Line 62: Instead of “Nrf2”, write “The protein Nrf2”. In general, authors should make sure that potential readers will understand their work, considering that readers may not be experts in the field. Another example (line 63): Instead of “Oltipraz”, write “Oltipraz, a synthetic dithiolethione”. Check throughout the manuscript.

-       Line 66: Instead of “natural products”, write “some natural products”

-       Line 72: Write “Auriculariales” in italics.

-       Line 79: add a reference.

Materials and methods

-       Line 90: add information about the ground size.

-       Line 114: instead of “DIO model”, write “Diet Induced Obesity (DIO) model”

-       Line 116: do authors mean “simvastatin” when they write “Sim”? again, authors should not give for granted that potential readers will understand every single term if they are not specified at least the first time they appear in the manuscript. Take this into account for the whole manuscript.

-       The exact and complete composition of all the diets used should be provided.

-       Did the authors make a sample size estimation for their experiment? If not, this should be added as a limitation of the study.

-       Lines 135-141: what authors mean when writing “#23227” or “#KT2827-A” and so on? Check the whole manuscript.

-       Lines 161-163: How authors decided about the number of cecal samples per group to be analyzed?

-       Regarding the statistical analysis: were all data normally distributed?  If not, how authors analyzed non-normally distributed data?

Conclusion

In my opinion the conclusion should be referred to the mice. Any reference to the possible impact on humans is too speculative. Taking this into account, I suggest to re-write the conclusion.
